# Oral and intravenous glucose administration elicit opposing microvascular blood flow responses in skeletal muscle of healthy people: role of incretins

Katherine M. Roberts-Thomson[1], Lewan Parker[1,2] , Andrew C. Betik[1,2] , Glenn D. Wadley[1] ,
Paul A. Della Gatta[1], Thomas H. Marwick[2] and Michelle A. Keske[1,2] 

[1] *Institute for Physical Activity and Nutrition (IPAN), School of Exercise and Nutrition Sciences Deakin University, Geelong, Victoria, Australia*
[2] *Baker Heart and Diabetes Institute, Melbourne, Victoria, Australia*

Edited by: Kim Barrett & Bettina Mittendorfer

Linked articles: This article is highlighted in a Perspective article by Tamariz-Ellemann *et al.* and a Journal Club article by Cohen & Wilkinson. To read these articles, visit https://doi.org/10.1113/JP282843 and https://doi.org/10.1113/JP282904.

The peer review history is available in the Supporting Information section of this article (https://doi.org/10.1113/JP282428#support-information-section).

**Abstract**   Insulin infusion increases skeletal muscle microvascular blood flow (MBF) in healthy people but is impaired during insulin resistance. However, we have shown that eliciting insulin secretion via oral glucose loading in healthy people impairs muscle MBF, whilst others have demonstrated intravenous glucose infusion stimulates MBF. We aimed to show that the route of glucose administration (oral *versus* intravenous) influences muscle MBF, and explore potential gut-derived hormones that may explain these divergent responses. Ten healthy individuals underwent a 120 min oral glucose tolerance test (OGTT; 75 g glucose) and on a subsequent occasion

Katherine M. Roberts-Thomson is a PhD candidate at the School of Exercise and Nutrition Sciences, Deakin University, Australia. Her PhD is focused on the impact of glucose on skeletal muscle vascular function in healthy and insulin-resistant populations. She received her Masters of nutrition and dietetics from the University of Sydney in 2014 and works as an accredited practising dietitian in the dietary management of type 2 diabetes. She is passionate about the translation of clinical research into dietary practice and management of chronic diseases.

The Journal of Physiology

an intravenous glucose tolerance test (IVGTT, bypassing the gut) matched for similar blood glucose excursions. Femoral artery and thigh muscle microvascular (contrast-enhanced ultrasound) haemodynamics were measured at baseline and during the OGTT/IVGTT. Plasma insulin, C-peptide, glucagon, non-esterified fatty acids and a range of gut-derived hormones and incretins (gastric inhibitory polypeptide (GIP) and glucagon-like peptide-1(GLP-1)) were measured at baseline and throughout the OGTT/IVGTT. The IVGTT increased whereas the OGTT impaired MBF (1.3-fold *versus* 0.5-fold from baseline, respectively, $P = 0.0006$). The impairment in MBF during the OGTT occurred despite producing 2.8-fold higher plasma insulin concentrations ($P = 0.0001$). The change in MBF from baseline ($\Delta$MBF) negatively correlated with $\Delta$GIP concentrations ($r = -0.665$, $P < 0.0001$). The natural log ratio of incretins GLP-1:GIP was positively associated with $\Delta$MBF ($r = 0.658$, $P < 0.0001$), suggesting they have opposing actions on the microvasculature. Postprandial hyperglycaemia *per se* does not acutely determine opposing microvascular responses between OGTT and IVGTT. Incretins may play a role in modulating skeletal muscle MBF in humans.

(Received 14 October 2021; accepted after revision 11 January 2022; first published online 18 January 2022)

**Corresponding author** M. A. Keske: Institute for Physical Activity and Nutrition (IPAN), School of Exercise and Nutrition Sciences, Deakin University, Geelong, VIC, Australia.　　Email: Michelle.Keske@deakin.edu.au

**Abstract figure legend** Oral versus intravenous glucose loading produces opposing effects on the skeletal muscle microvasculature. The release of glucose-dependent insulinotropic polypeptide (GIP) may be involved in the impaired microvascular responses to oral glucose loading.

## Key points

- Insulin or mixed nutrient meals stimulate skeletal muscle microvascular blood flow (MBF) to aid in the delivery of nutrients; however, this vascular effect is lost during insulin resistance.
- Food/drinks containing large glucose loads impair MBF in healthy people; however, this impairment is not observed when glucose is infused intravenously (bypassing the gut).
- We investigated skeletal muscle MBF responses to a 75 g oral glucose tolerance test and intravenous glucose infusion and aimed to identify potential gut hormones responsible for glucose-mediated changes in MBF.
- Despite similar blood glucose concentrations, orally ingested glucose impaired, whereas intravenously infused glucose augmented, skeletal muscle MBF. The incretin gastric inhibitory polypeptide was negatively associated with MBF, suggestive of an incretin-mediated MBF response to oral glucose ingestion.
- This work provides new insight into why diets high in glucose may be detrimental to vascular health and provides new avenues for novel treatment strategies targeting microvascular dysfunction.

## Introduction

Skeletal muscle microvascular blood flow (MBF) increases in response to insulin infusion (hyperinsulinaemic euglycaemic clamp) (Coggins *et al.* 2001; Eggleston *et al.* 2007; Jahn *et al.* 2016) or a mixed-nutrient meal (Vincent *et al.* 2006; Keske *et al.* 2009; Russell *et al.* 2018). This increase in skeletal muscle MBF promotes nutrient and hormone delivery to the myocyte (reviewed in (Clark, 2008; Wagenmakers, 2016)) and contributes to 40–50% of insulin-stimulated skeletal muscle glucose disposal (Vincent *et al.* 2003; Vincent *et al.* 2004). This favourable vascular response is blunted in individuals that are obese (Clerk *et al.*

2006; Keske *et al.* 2009; Meijer *et al.* 2015; Wang *et al.* 2020), have metabolic syndrome (Jahn *et al.* 2016), have a parent with type 2 diabetes (Russell *et al.* 2022) or themselves have type 2 diabetes (Emanuel *et al.* 2018), positioning the skeletal muscle microvasculature as an important physiological tissue for human metabolism.

Insulin infusion (hyperinsulinaemic euglycaemic clamp) is commonly used to mimic postprandial metabolic and vascular responses (DeFronzo *et al.* 1979; Baron & Brechtel, 1993). However, a growing body of work suggests that the true postprandial state (following the ingestion of food/beverages) is more complex than intravenous infusions (Roberts-Thomson et al. 2020), with dynamic changes in blood glucose,

amino acids and insulin occurring, which are largely absent during insulin infusion (Spiller *et al.* 1987; Vincent *et al.* 2006; Roberts-Thomson et al. 2020). One important variation between these conditions is the route via which glucose/nutrients are administered (orally *versus* intravenously). Oral administration of glucose elicits a significantly greater insulin response than an equivalent dose administered intravenously (Elrick *et al.* 1964). The 'incretin effect' refers to the secretion of the insulin-promoting gut hormones glucagon-like peptide-1 (GLP-1) and gastric inhibitory polypeptide (GIP) from the K-cells of the small intestine following nutrient or food ingestion (Baggio & Drucker, 2007). In addition to their insulinotropic properties, there is evidence of their vasoactivity (Baggio & Drucker, 2007; Sjøberg *et al.* 2013; Chai *et al.* 2014; Subaran *et al.* 2014; Tan *et al.* 2018; Wang *et al.* 2020). GLP-1 infusion has been demonstrated to increase microvascular perfusion in the skeletal muscle of animals (Sjøberg *et al.* 2013; Chai *et al.* 2014) and humans (Sjøberg *et al.* 2013; Subaran *et al.* 2014; Tan *et al.* 2018; Wang *et al.* 2020), supporting GLP-1 as a key regulator of postprandial vascular function and metabolism. However, the effects of other glucoregulatory gut hormones such as GIP, peptide YY (PYY) and ghrelin, on skeletal muscle MBF in the postprandial state are not known.

Although it is well accepted that insulin stimulates muscle MBF, the ingestion of 50–75 g of glucose (on its own or in combination with a mixed nutrient meal) either impairs (Russell *et al.* 2018; Parker *et al.* 2020) or does not alter (Tobin *et al.* 2010) MBF in skeletal muscle of healthy individuals, despite physiological increases in plasma insulin levels. There are well-known mechanisms that may explain why acute hyperglycaemia impairs vascular function, such as impaired nitric oxide (NO) signalling in the endothelium and augmented production of endothelium-derived vasoconstrictors such as prostanoids (Tesfamariam *et al.* 1990; Renaudin *et al.* 1998; De Nigris *et al.* 2015). Therefore, it can be hypothesised that the extent of the glucose excursion in the circulation is linked to the extent of the impairment in MBF in skeletal muscle (Russell *et al.* 2018; Parker *et al.* 2020). However, others have demonstrated that when glucose is administered intravenously into healthy humans (Horton *et al.* 2020) or non-human primates (Chadderdon *et al.* 2012, 2016), skeletal muscle MBF increases. As such, the route of glucose administration (i.e. glucose challenge with/without gut involvement), rather than hyperglycaemia in the circulation itself, may be the key factor behind divergent skeletal muscle MBF responses. The aim of the current study is to compare the vascular effects of glucose when administered orally *versus* intravenously when matched for blood glucose excursions in healthy humans. It was hypothesised that intravenously infused glucose would stimulate skeletal muscle MBF whereas oral glucose ingestion would impair MBF. Furthermore, it was hypothesised that increases in circulating gut-derived hormones would be linked to oral glucose-induced impairments in skeletal muscle MBF.

Parts of this study were published in abstract form at the 80th American Diabetes Association Scientific Meeting, June 2020.

## Methods

### General

This study was approved by the Deakin University Human Research Ethics Committee (2019-062) and conformed to the standards set by the *Declaration of Helsinki*. All participants provided written informed consent. Participants were included in the study if they were aged between 18 and 50 years, were normal to overweight (BMI range: 18.5–30.0 kg/m$^2$) and normotensive (seated brachial blood pressure: <140/90 mmHg). Individuals were excluded if they were outside the age and BMI range, had a first degree relative or >1 grandparent with diagnosed type 2 diabetes, hypertensive (blood pressure: >140/90 mmHg), current smoker, pregnant or lactating, or had any personal history of cardiovascular disease, malignancy, diabetes, liver disease, pulmonary disease, arthritis or musculoskeletal disease. Based on an *a priori* sample size calculation, it was estimated that 13 participants would be required to detect a two-fold difference in MBF between the oral glucose tolerance test (OGTT) and intravenous glucose tolerance test (IVGTT) (SD = 115%, effect size = 0.87, $\alpha = 0.05$, power = 80%) based on previous work (Russell *et al.* 2018). Ten participants completed the study between September 2019 and March 2020 prior to COVID-19-related clinic closures.

### Participant screening visit

All participants attended a brief screening visit at the testing facility where body weight, height and blood pressure were assessed, and a medical questionnaire completed, to confirm eligibility. Participants were scheduled for two clinical testing visits between 1 and 4 weeks apart. In order to match blood glucose excursions between the testing visits, the OGTT was performed on the first visit and the variable rate IVGTT on the second clinical testing visit. This design prohibited treatment randomisation and blinding. Ten participants completed both clinical testing visits of the study (OGTT and IVGTT).

### Clinical testing visits

Participants were fasted overnight for 12 h and refrained from exercise and alcohol for 48 h prior to each clinical

**Table 1. Glucose infusion rates (10% glucose w/v) over the 120 min IVGTT**

| Time (minutes) | Infusion rate (ml/min) |
|---|---|
| 0 | 1.5 ± 1.7 |
| 5 | 1.3 ± 1.5 |
| 10 | 3.3 ± 2.0 |
| 15 | 4.4 ± 1.4 |
| 20 | 5.0 ± 1.6 |
| 30 | 4.5 ± 1.4 |
| 40 | 2.9 ± 2.4 |
| 50 | 2.5 ± 3.3 |
| 60 | 1.7 ± 2.0 |
| 70 | 2.1 ± 2.1 |
| 80 | 2.3 ± 2.4 |
| 90 | 1.6 ± 2.3 |
| 100 | 1.3 ± 2.4 |
| 110 | 1.5 ± 2.5 |
| 120 | 0.0 ± 0.0 |
| Total glucose infused (g) | 29.9 ± 17.6 |

Data are expressed as means ± SD for $n$ = 10 participants.

testing visit. A catheter was placed into the antecubital vein of one arm for blood draws and microbubble infusion for the OGTT testing visit, and an additional catheter placed into the antecubital vein of the alternate arm for the variable rate IVGTT.

**OGTT.** Participants consumed a 75 g glucose drink (Fronine, Carbotest, Thermo Fisher Scientific, Scoresby, Australia). Venous blood was sampled at 0, 5, 10, 15, 20, 30, 40, 50, 60, 80, 90, 100, 110 and 120 min following glucose ingestion.

**Variable rate IVGTT.** Participants underwent a variable rate intravenous glucose infusion (glucose 10% w/v solution, Baxter Healthcare, Brunswick, Australia) to mimic the blood glucose excursions elicited by the OGTT. Venous blood was sampled at the same time points as the OGTT. The glucose infusion rate was adjusted accordingly over the 120 min time course to match the glucose levels observed during the OGTT for each participant (Table 1).

### Skeletal muscle microvascular perfusion

Contrast-enhanced ultrasound was used to measure skeletal muscle microvascular perfusion in the quadriceps of the right leg. A linear array (L9-3) ultrasound transducer interfaced with an ultrasound machine (iU22; Philips Healthcare, Melbourne, Australia) was used as described previously (Russell *et al.* 2018; Parker *et al.* 2020). An intravenous contrast agent containing

echogenic microbubbles (DEFINITY®, Lantheus Medical Imagining, Keilor Park, Australia) was diluted into saline solution (1 ml into 30 ml saline), and continuously infused (2.0–2.6 ml/min, based on the participant's body weight and the degree of tissue opacification) for skeletal muscle imaging. Once the systemic concentration of microbubbles reached steady state (∼5 min), a high-energy pulse of ultrasound (mechanical index 1.3) was transmitted to destroy microbubbles in the selected region of imaging (vastus lateralis). Rate of reperfusion (mechanical index 0.11) into the microvasculature of the muscle was measured in real-time at baseline and repeated at 60 and 120 min during the OGTT/IVGTT. Three data captures of 45 s were collected at each time point and averaged together and analysed using QLAB software (Philips Healthcare, Melbourne, Australia). All images were background subtracted (0.5 s image) to eliminate signal from larger fast filling blood vessels and tissue *per se*. Background-subtracted acoustic intensity *vs.* time was fitted to the function: $y = A(1 - e^{-\beta(t - {}^t\mathrm{b})})$, where $y$ is the acoustic intensity at time $t$, $t_\mathrm{b}$ is the background time, $A$ is the plateau of acoustic-intensity (microvascular blood volume; MBV) and $\beta$ is the rate constant (a measure of microvascular refilling rate), as previously described (Russell *et al.* 2017, 2018). MBF was calculated using $A \times \beta$. Image analysis was performed identically at baseline, 60 and 120 min post-OGTT/IVGTT.

### Superficial femoral and large artery haemodynamics

Diameter, blood velocity and flow measurements of the superficial femoral artery were performed using a linear array (L12-5) ultrasound transducer interfaced with an ultrasound machine (iU22; Philips Healthcare, Melbourne, Australia). Femoral artery diameter was measured in triplicate at the same phase of the cardiac cycle (R-wave, based on the QRS complex) using 2D imaging of the longitudinal artery. Femoral artery velocity was assessed using pulse-wave Doppler quantified by an automated tracing software and averaged over ∼10 heartbeats. Femoral artery blood flow (ml/min) was calculated using $\pi r^2 \times$ mean velocity $\times$ 60, where $r$ is radius (cm) and mean velocity measured in cm/s.

All participants were fitted with a Mobil-O-Graph monitor validated to measure central and brachial blood pressure, heart rate, mean arterial pressure, augmentation index and pulse wave velocity (I.E.M., Stolberg, Germany). Measures were performed in triplicate at baseline, 60 and 120 min during the OGTT/IVGTT.

### Blood sampling and plasma analysis

A fasting blood sample was taken during clinic visit 1 and sent to a nationally accredited pathology laboratory

(Australian Clinical Labs, Burwood, Australia) for analysis of clinical chemistry including fasting glucose, glycosylated haemoglobin (HbA$_{1c}$), total cholesterol, low-density lipoprotein cholesterol (LDL), high-density lipoprotein cholesterol (HDL), and triglycerides. Blood glucose was measured throughout the time course of the OGTT and IVGTT clinical testing visits using an automated radiometer analysis system (ABL800 FLEX; Radiometer Medical, Copenhagen, Denmark). Blood sampled throughout the time course of the OGTT/IVGTT was collected in BD 800 Vacutainers (Becton Dickinson, Mulgrave, Australia) containing protease, esterase and dipeptidyl peptidase-4 inhibitors for preservation of GLP-1, GIP and other gut-derived hormones.

Plasma insulin, C-peptide and active GLP-1 were determined using an enzyme-linked immunosorbent assay (ELISA) (ALPCO Diagnostics, Windham, NH, USA). Plasma non-esterified fatty acids (NEFA) levels were determined using an enzymatic colorimetric assay (Wako Pure Chemical Industries, Osaka, Japan). Plasma total GLP-1, total GIP, active ghrelin, PYY and glucagon were determined using a Milliplex multiplex assay (Millipore Human Metabolic Hormone Panel, Merck, Bayswater, Australia).

### Statistical analysis

All statistical analysis was undertaken using GraphPad Prism (version 8.0, GraphPad Software, La Jolla, CA, USA) except for correlations, which were performed using SigmaPlot (Systat Software, San Jose, CA, USA). All data are expressed as means $\pm$ SD. Two-way repeated measures ANOVA was used to compare multiple means with time (at baseline and throughout the 120 min OGTT/IVGTT time course) and condition (OGTT *versus* IVGTT) as within-subject factors. Mixed model analyses were used for statistical testing of central and peripheral haemodynamic measures due to missing data for one participant. All data that were not normally distributed were transformed using natural log. Significant inter-action and main effects were explored *post hoc* using Fisher's least significant difference test. Associations between variables (gut-derived hormones) and muscle MBV, $\beta$ and MBF were studied with the use of Pearson's correlations. Correlations were performed on changes in MBV, $\beta$ and MBF *versus* changes in active GLP-1, total GLP-1, total GIP, natural log total GLP:GIP, ghrelin and PYY. Data were initially correlated with combined time-point data from the OGTT and IVGTT (both 60 and 120 min). If a statistically significant correlation with an *r*-value >0.4 was found, correlations were then performed on individual time points (60 and 120 min). All statistical analysis was conducted at 95% level of significance ($P < 0.05$).

## Table 2. Participant characteristics

|  | Mean $\pm$ SD | Range |
|---|---|---|
| Age (years) | 29 $\pm$ 8 | 20–47 |
| Sex (M/F) | 4/6 | — |
| Weight (kg) | 71.0 $\pm$ 14.4 | 53.4–98.9 |
| Height (cm) | 172.7 $\pm$ 10.3 | 158.2–189.0 |
| BMI (kg/m$^2$) | 23.6 $\pm$ 3.0 | 20.6–29.4 |
| Fasting plasma glucose (mmol/l) | 4.5 $\pm$ 0.3 | 4.0–5.1 |
| HbA$_{1c}$ (%) | 5.2 $\pm$ 0.3 | 4.8–5.6 |
| HbA$_{1c}$ (mmol/mol) | 33.2 $\pm$ 3.0 | 29.0–38.0 |
| Fasting plasma insulin (pmol/l) | 34.6 $\pm$ 11.6 | 18.1–59.1 |
| Fasting plasma lipids |  |  |
| Total cholesterol (mmol/l) | 4.3 $\pm$ 1.4 | 2.6–7.5 |
| LDL (mmol/l) | 2.4 $\pm$ 1.2 | 1.1–5.1 |
| HDL (mmol/l) | 1.5 $\pm$ 0.3 | 1.1–1.9 |
| Triglycerides (mmol/l) | 0.9 $\pm$ 0.3 | 0.7–1.4 |
| Resting brachial blood pressure |  |  |
| SBP (mmHg) | 114 $\pm$ 9 | 104–130 |
| DBP (mmHg) | 74 $\pm$ 9 | 60–87 |

Data are expressed as means $\pm$ SD and range for $n = 10$ participants. BMI, body mass index; DBP, diastolic blood pressure; HDL, high-density lipoprotein cholesterol; LDL, low-density lipoprotein cholesterol; SBP, systolic blood pressure.

## Results

### Participant characteristics

Participant characteristics and anthropometrics are reported in Table 2. Participants were aged between 20 and 47 years (29 $\pm$ 8 years). Participants were apparently healthy with normal fasting blood glucose (<6.5 mmol/l), HbA$_{1C}$ (<6.0% and <42 mmol/mol), plasma insulin levels (<174 pmol/l) and seated brachial blood pressure (<140/90 mmHg).

### Metabolic responses

Blood glucose levels increased significantly at 15–120 min during the OGTT ($P = 0.024$ to 0.001, respectively) and at 20–120 min during the IVGTT ($P = 0.003$ to 0.018, respectively, Fig. 1*A*). The glucose area under the time curve (AUC) was similar between the OGTT and IVGTT (799.3 $\pm$ 132.4 *versus* 804.2 $\pm$ 123.3 mmol/l$\times$120 min, $P = 0.703$, Fig. 1*B*).

Despite plasma insulin concentrations increasing over time during both conditions, the OGTT was significantly higher compared to the IVGTT from 20 to 120 min ($P = 0.0196$ to 0.0001, respectively, Fig. 1*C*). Plasma insulin AUC was significantly higher during the OGTT compared to the IVGTT (26094.7 $\pm$ 11087.9 *versus* 9237.0 $\pm$ 3757.5 pmol/l$\times$120 min, $P = 0.0001$, Fig. 1*D*). These effects were similar for plasma C-peptide (Fig. 1*E* and *F*).

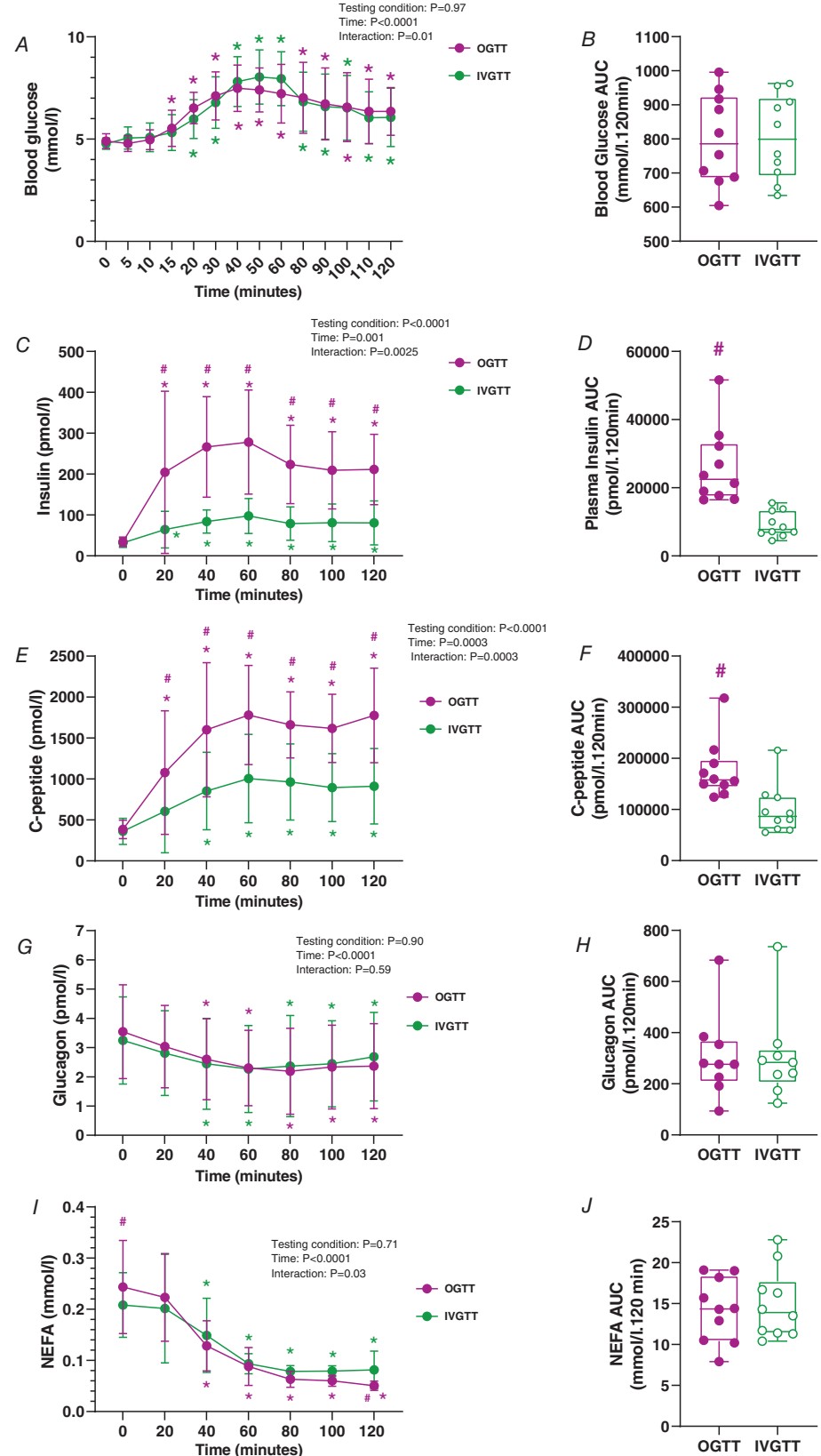

**Figure 1. Metabolic effects of an OGTT and IVGTT**
Graphs show OGTT (magenta) and IVGTT (green). *A* and *B*, 2 h blood glucose time course (*A*) and blood glucose area under the time curve (AUC) (*B*). *C* and *D*, 2 h plasma insulin time course (*C*) and plasma insulin AUC (*D*).

*E* and *F*, 2 h plasma C-peptide time course (*E*) and plasma C-peptide AUC (*F*). *G* and *H*, 2 h plasma glucagon time course (*G*) and plasma glucagon AUC (*H*). *I* and *J*, 2 h plasma non-esterified fatty acid (NEFA) time course (*I*) and plasma NEFA AUC (*J*). For visual clarity, time-course data are expressed as line graphs with means ± SD. Bar graphs are presented as box and whisker plots with individual data points. The box represents the interquartile range alongside the median (line). The whiskers represent the minimum and maximum range of data for *n* = 10 participants. *$P < 0.05$ *versus* 0 min, #$P < 0.05$ *versus* IVGTT for the same time point (refer to Results for exact *P*-values). [Colour figure can be viewed at wileyonlinelibrary.com]

**Table 3. Central and peripheral haemodynamics following the OGTT and IVGTT**

| | bSys BP (mmHg) | bDias BP (mmHg) | cSys BP (mmHg) | cDias BP (mmHg) | HR (BPM) | Aug Index (adjusted 75 BPM) | MAP (mmHg) | PWV (m/s) |
|---|---|---|---|---|---|---|---|---|
| OGTT | | | | | | | | |
| 0 min | 114 ± 9 | 74 ± 9 | 105 ± 9 | 75 ± 9 | 61 ± 7 | 13 ± 7 | 93 ± 9 | 5 ± 1 |
| 60 min | 115 ± 12 | 71 ± 5 | 105 ± 11 | 72 ± 6 | 59 ± 9 | 18 ± 15 | 91 ± 8 | 5 ± 1 |
| 120 min | 113 ± 11 | 71 ± 9 | 102 ± 9 | 72 ± 10 | 61 ± 10 | 9 ± 7 | 91 ± 9 | 5 ± 1 |
| IVGTT | | | | | | | | |
| 0 min | 119 ± 12 | 76 ± 11 | 107 ± 14 | 74 ± 10 | 64 ± 7 | 6 ± 8 | 97 ± 12 | 5 ± 1 |
| 60 min | 118 ± 17 | 73 ± 11 | 105 ± 14 | 69 ± 7 | 61 ± 9 | 9 ± 12 | 94 ± 12 | 5 ± 1 |
| 120 min | 113 ± 11 | 78 ± 10 | 102 ± 8 | 75 ± 8 | 60 ± 10 | 9 ± 9 | 94 ± 10 | 5 ± 1 |

Table shows haemodynamic measures at 0, 60 and 120 min during an OGTT and IVGTT. Data are expressed as means ± SD. *n* = 9 participants. Aug Index, augmentation index; bDias BP, brachial diastolic blood pressure; bSys BP, brachial systolic blood pressure; cDias BP, central diastolic blood pressure; cSys BP, central systolic blood pressure; HR, heart rate; MAP, mean arterial pressure; PWV, pulse wave velocity.

Plasma glucagon decreased significantly at 40–120 min compared to 0 min irrespective of the treatment (OGTT: $P = 0.0189$ to 0.0203; IVGTT: $P = 0.0119$ to 0.0298, respectively, Fig. 1*G*). There were no differences in plasma glucagon AUC between treatments ($P = 0.9211$, Fig. 1*H*).

Plasma NEFA concentrations decreased at 40–120 min compared to 0 min during both treatments (OGTT: $P < 0.0001$ to <0.0001; IVGTT: $P = 0.0002$ to <0.0001, respectively, Fig. 1*I*). Plasma NEFA AUC was similar between treatments ($P = 0.6569$, Fig. 1*J*).

### Skeletal muscle microvascular measures

There were no interactions ($P = 0.09$) or effect of treatment ($P = 0.61$) or time ($P = 0.64$) for skeletal muscle MBV (Fig. 2*A*) or ΔMBV ($P = 0.44$, 0.30 and 0.73, respectively) for both groups (Fig. 2*B*).

Skeletal muscle $\beta$ (microvascular flow velocity) significantly decreased at 60 min compared to 0 min during the OGTT ($P = 0.0321$); however, it was significantly higher at 60 min during the IVGTT ($P = 0.0223$). These effects were absent at 120 min (OGTT: $P = 0.0806$; IVGTT: $P = 0.9622$, Fig. 2*C*). However, the $\Delta\beta$ was significantly lower during the OGTT compared to the IVGTT at both 60 ($P = 0.0073$) and 120 min ($P = 0.0498$, Fig. 2*D*).

MBF was lower at 60 and 120 min during the OGTT compared to 0 min ($P = 0.0077$ and 0.0009, respectively). In contrast, MBF was higher at 60 min compared to 0 min during the IVGTT ($P = 0.0133$); however, it was unchanged at 120 min ($P = 0.1667$, Fig. 2*E*). Skeletal muscle MBF was lower during the OGTT compared to the IVGTT at both 60 and 120 min ($P = 0.0163$ and 0.0085, respectively). ΔMBF was lower at both 60 and 120 min during the OGTT compared to the IVGTT ($P = 0.0158$ and 0.0005, respectively, Fig. 2*F*).

### Large artery haemodynamic responses

No changes in superficial femoral artery diameter, velocity or blood flow were detected (data not shown).

Central and peripheral haemodynamic measures at baseline, 60 and 120 min during the OGTT and IVGTT are reported in Table 3. No interactions were noted for central blood pressure (central systolic blood pressure: $P = 0.8945$, central diastolic blood pressure: $P = 0.3529$), brachial blood pressure (brachial systolic blood pressure: $P = 0.3870$; brachial diastolic blood pressure: $P = 0.4330$), heart rate ($P = 0.7069$), augmentation index ($P = 0.1395$), pulse wave velocity ($P = 0.7059$) or mean arterial pressure ($P = 0.7206$). Due to equipment malfunction, Mobil-O-Graph data from one participant were lost.

### Plasma incretin responses

Plasma incretin responses to the OGTT and IVGTT are reported in Fig. 3. Active GLP-1 increased at 20–60 min

($P < 0.0001$ to 0.0003, respectively) and 100–120 min ($P = 0.0106$ to 0.0005, respectively) during the OGTT, compared to 0 min. Active GLP-1 did not change during the IVGTT compared to 0 min except for a small decrease at 80 min ($P = 0.0348$, Fig. 3*A*). Active GLP-1 was significantly higher during the OGTT compared to IVGTT at 20–80 min ($P < 0.0001$ to <0.0001, respectively) and at 120 min ($P = 0.0049$, Fig. 3*A*). As such, $\Delta$ active GLP-1 was significantly higher from 20–80 min during the OGTT compared to IVGTT ($P < 0.0001$ to 0.0043, respectively, Fig. 3*B*).

Total plasma GLP-1 increased at 20 and 40 min from 0 min during the OGTT ($P < 0.0001$ for both). In contrast, total GLP-1 decreased from 40–120 min during the IVGTT compared to 0 min ($P < 0.0001$ to <0.0001, respectively, Fig. 3*C*). Total GLP-1 levels were significantly higher at all time points during the OGTT, compared to IVGTT ($P < 0.0001$ to 0.0020, respectively, Fig. 3*C*). The OGTT elicited a larger $\Delta$ total GLP-1 at all time points compared to the IVGTT ($P < 0.0001$ to 0.0024, Fig. 3*D*).

Total GIP concentrations increased at all time points during the OGTT compared to 0 min (all $P < 0.0001$), with levels significantly greater than the IVGTT at all time points (all $P < 0.0001$, Fig. 3*E*). Total GIP remained unchanged during the time course of the IVGTT. This was also reflected by a higher $\Delta$ total GIP during the

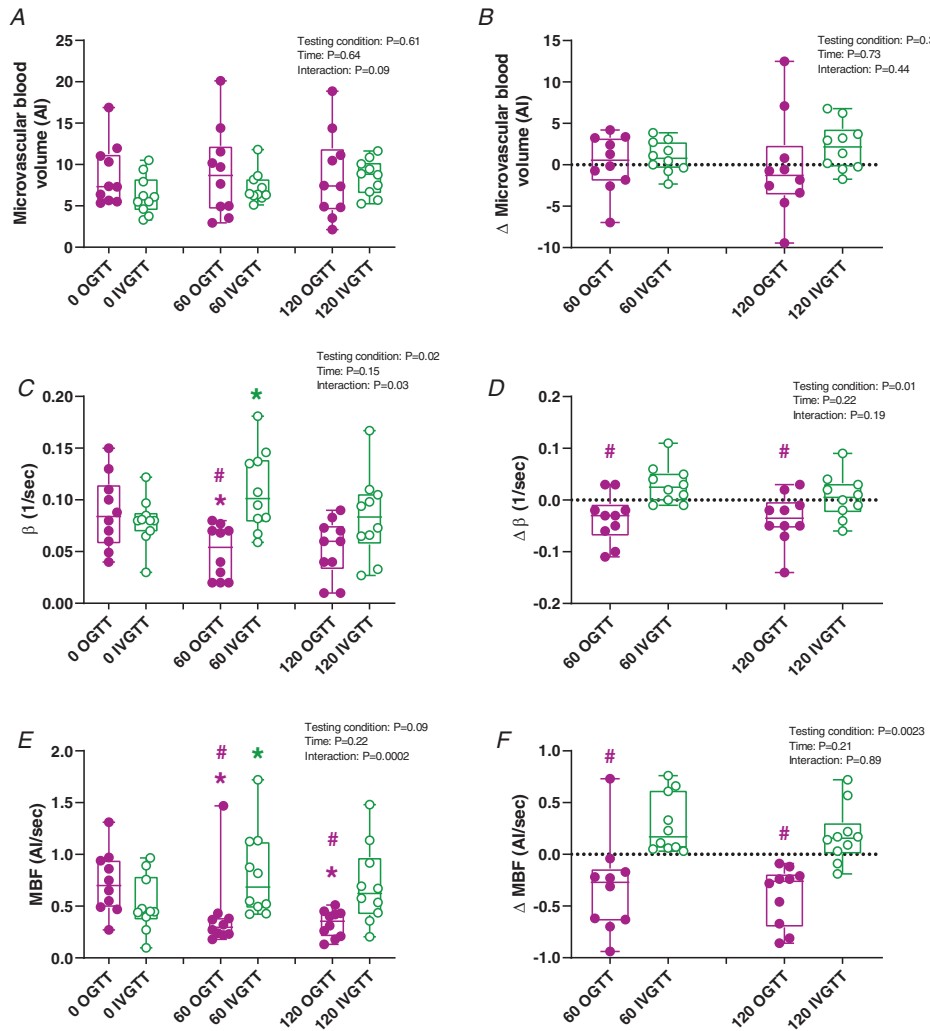

**Figure 2. Skeletal muscle microvascular responses to an OGTT and IVGTT**
Graphs show OGTT (magenta) and IVGTT (green). *A* and *B*, skeletal muscle microvascular blood volume (MBV) at 0, 60 and 120 min (*A*), and $\Delta$MBV at 60 and 120 min (*B*) from baseline. *C* and *D*, skeletal muscle microvascular $\beta$ at 0, 60 and 120 min (*C*), and $\Delta\beta$ at 60 and 120 min (*D*) from baseline. *E* and *F*, skeletal muscle microvascular blood flow (MBF) at 0, 60 and 120 min (*E*), and $\Delta$MBF at 60 and 120 min (*F*) from baseline. Bar graphs are presented as box and whisker plots with individual data points. The box represents the interquartile range alongside the median (line). The whiskers represent the minimum and maximum range of data for $n = 10$ participants. *$P < 0.05$ *versus* 0 min, #$P < 0.05$ *versus* IVGTT for the same time point (refer to Results for exact $P$-values). AI, Acoustic Intensity. [Colour figure can be viewed at wileyonlinelibrary.com]

OGTT than the IVGTT at all time points (all $P < 0.0001$, Fig. 3*F*).

The ratio of total GLP-1:GIP was significantly lower during the OGTT at all time points, compared to 0 min (all $P < 0.0001$). Furthermore, this ratio was lower during the OGTT compared to the IVGTT at all time points ($P < 0.0001$, Fig. 3*G*). The GLP-1:GIP ratio remained unchanged during the time course of the IVGTT.

## Other gut hormone responses

Gut hormone responses to the OGTT and IVGTT are shown in Table 4. Active ghrelin was lower at 20 and 40 min during the OGTT ($P = 0.0281$ and 0.0160, respectively), compared to the IVGTT. A small but significant increase in plasma PYY was observed at 20 min during the OGTT compared to 0 min ($P = 0.0278$), but

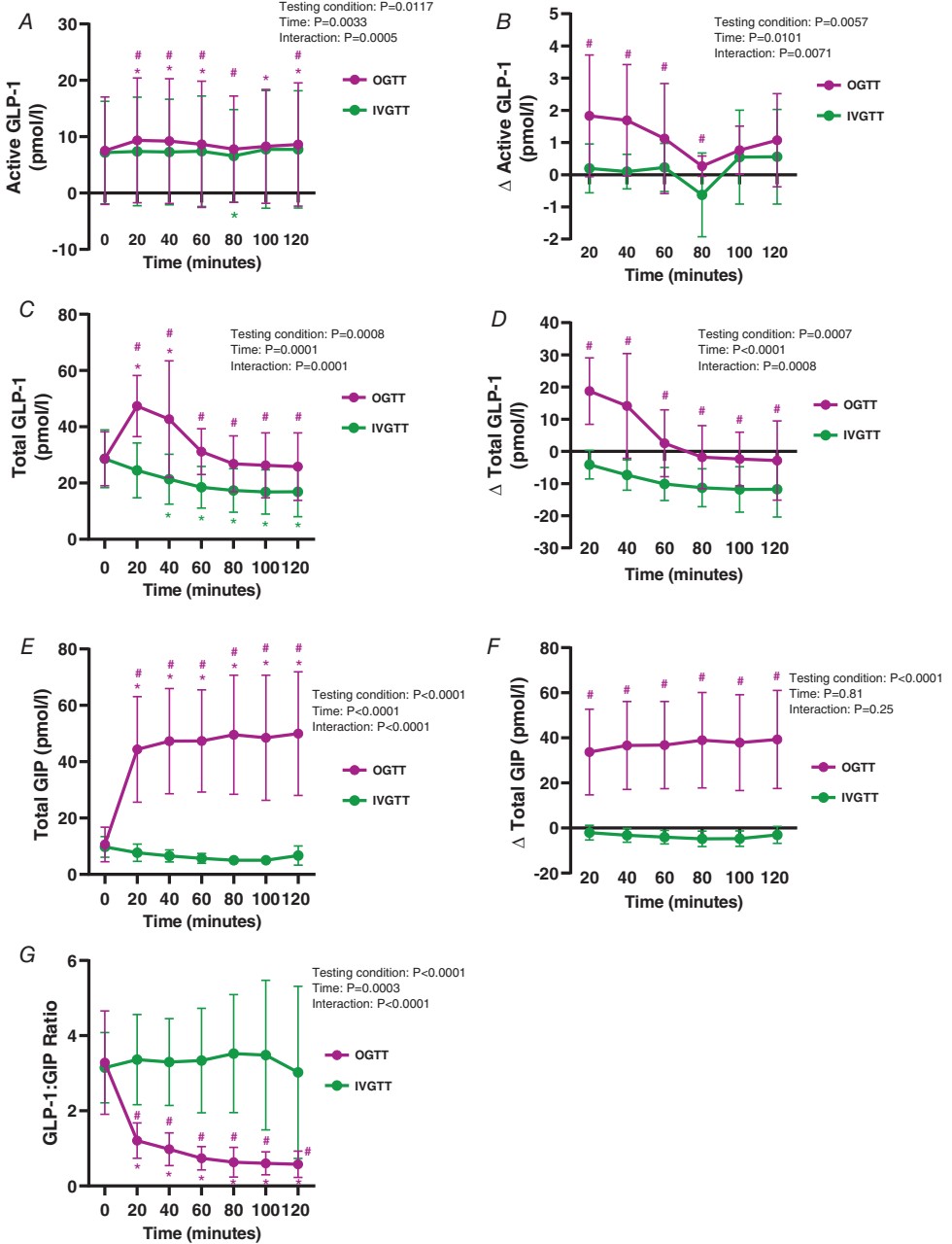

**Figure 3. Plasma incretin responses to an OGTT and IVGTT**
Graphs show OGTT (magenta) and IVGTT (green). *A* and *B*, 2 h active GLP-1 time course (*A*) and Δ active GLP-1 time course (*B*). *C* and *D*, 2 h total GLP-1 time course (*C*) and Δ total GLP-1 time course (*D*). *E* and *F*, 2 h total GIP time course (*E*) and Δ total GIP time course (*F*). *G*, 2 h GLP-1:GIP ratio time course. For visual clarity, time-course data are expressed as line graphs with means ± SD for *n* = 10 participants. *$P < 0.05$ *versus* 0 min, #$P < 0.05$ *versus* IVGTT for the same time point (refer to Results for exact *P*-values). [Colour figure can be viewed at wileyonlinelibrary.com]

**Table 4. Gut hormone responses to an OGTT and IVGTT**

| | Time (min) | | | | | | |
|---|---|---|---|---|---|---|---|
| | **0** | **20** | **40** | **60** | **80** | **100** | **120** |
| Active ghrelin (pmol/l) | | | | | | | |
| OGTT | 13.7 ± 12.3 | 8.8 ± 6.7[#] | 6.8 ± 3.8[#] | 9.1 ± 7.4 | 9.2 ± 7.1 | 11.6 ± 9.6 | 11.3 ± 7.7 |
| OGTT – Δ from 0 min | — | −4.9 ± 8.2 | −6.9 ± 9.3 | −4.7 ± 9.7 | −4.5 ± 7.2 | −2.1 ± 5.0 | −2.5 ± 6.3 |
| IVGTT | 13.4 ± 9.9 | 14.8 ± 11.1 | 15.8 ± 10.9 | 10.3 ± 8.2 | 11.7 ± 6.7 | 16.3 ± 9.0 | 17.3 ± 10.3 |
| IVGTT – Δ from 0 min | — | 1.4 ± 9.1 | 2.4 ± 7.0 | −3.0 ± 10.2 | −1.7 ± 5.7 | 2.9 ± 6.4 | 3.9 ± 8.9 |
| PYY (pmol/l) | | | | | | | |
| OGTT | 24.5 ± 18.3 | 26.7 ± 17.5*[#] | 25.1 ± 17.2[#] | 24.9 ± 17.8[#] | 23.8 ± 18.1 | 24.7 ± 18.0 | 23.7 ± 18.6 |
| OGTT – Δ from 0 min | — | 2.1 ± 2.2[#] | 0.6 ± 2.2[#] | 0.4 ± 2.4[#] | −0.7 ± 4.0[#] | 0.2 ± 2.2[#] | −0.8 ± 3.0[#] |
| IVGTT | 22.9 ± 18.7 | 20.7 ± 19.6 | 20.3 ± 19.8 | 20.3 ± 20.0* | 19.8 ± 19.9 | 20.2 ± 20.5 | 19.9 ± 20.0* |
| IVGTT – Δ from 0 min | — | −2.2 ± 3.1 | −2.7 ± 3.3 | −2.6 ± 2.9 | −3.1 ± 4.1 | −2.7 ± 3.3 | −3.1 ± 2.7 |

Hormone levels over 120 min and Δ from baseline. Data are expressed as means ± SD for *n* = 10 participants. *$P < 0.05$ *versus* 0 min, [#]$P < 0.05$ *versus* IVGTT for the same time point (refer to Results for exact *P*-values). PYY, peptide YY.

not at any other time point. Plasma PYY was significantly lower at 60 and 120 min compared to 0 min during the IVGTT ($P = 0.0397$ and 0.0143, respectively). PYY levels were significantly higher at 20 ($P = 0.0103$), 40 ($P = 0.0190$), and 60 min ($P = 0.0250$) during the OGTT compared to IVGTT (Table 4). ΔPYY was significantly higher at all time points during the OGTT compared to the IVGTT (Table 4).

## Associations between gut-derived hormones and skeletal muscle microvascular blood flow

Figure 4 shows significant associations between changes in microvascular responses and changes in plasma gut-derived hormones during the OGTT and IVGTT with an *r*-value >0.4. ΔMBF was negatively associated with ΔGIP ($r = -0.665$; $P < 0.0001$) and this relationship was stronger at 120 min ($r = -0.788$; $P < 0.0001$) compared to 60 min ($r = -0.566$; $P = 0.00934$, Fig. 4A). These effects were also similar for Δβ with ΔGIP (Fig. 4B).

In contrast, ΔMBF was positively associated with natural log GLP-1:GIP ratio ($r = 0.658$; $P < 0.0001$) and this relationship was stronger at 120 min ($r = 0.726$; $P = 0.0003$) compared to 60 min ($r = 0.601$; $P = 0.0051$, Fig. 4C). These effects were similar for Δβ and natural log GLP-1:GIP ratio (Fig. 4D).

There were no associations between MBV and any gut hormone measured in this study. There were no associations between MBF and active GLP-1, total GLP-1, ghrelin or PYY.

## Discussion

This is the first study to directly compare the peripheral macro- and microvascular effects of oral *versus* intravenous glucose loading in healthy humans. We

demonstrate that (1) skeletal muscle MBF decreases with an OGTT and increases with an IVGTT when matched for similar blood glucose excursions; (2) these opposing vascular actions are not related to hyperglycaemia *per se*; (3) plasma levels of the incretin hormone GIP are strongly associated with impaired microvascular responses in skeletal muscle, whereas (4) plasma PYY and ghrelin are not associated with MBF. These findings suggest a novel incretin-mediated mechanism regulating skeletal muscle MBF responses following oral glucose loading.

The effects of acute hyperglycaemia on impaired vascular actions have been investigated in endothelial cell culture/in vitro models (Tesfamariam *et al.* 1990; De Nigris et al. 2015), animal models (Renaudin *et al.* 1998; 1999) and humans (Giugliano *et al.* 1997; Williams *et al.* 1998). We previously proposed that the extent of hyperglycaemia following nutrient ingestion is directly related to skeletal muscle MBF responses (Russell *et al.* 2018), and a greater postprandial glucose AUC is associated with a greater decrease in muscle MBF (Parker *et al.* 2020). However, prior work shows intravenously infused glucose (alone or in combination with insulin) increases skeletal muscle microvascular responses in healthy non-human primates (Chadderdon *et al.* 2012) and humans (Horton *et al.* 2020). We now provide evidence that the route of glucose administration (glucose loading with/without involvement of the gut), and not the presence of hyperglycaemia in the blood, is the key determinant of divergent responses in muscle MBF between oral and intravenous glucose loading. Despite being matched for similar blood glucose concentrations, MBF is blunted during orally ingested glucose (OGTT) but stimulated during intravenously administered glucose (IVGTT). We postulate that glucose ingestion stimulates incretin secretion from the gut and this inhibits vasodilatation leading to impaired skeletal muscle MBF. We do not know why the OGTT would restrict flow to skeletal muscle in

healthy individuals; however, blood flow redistribution (and therefore altered nutrient disposal) to other insulin sensitive tissues is a possibility. We (Hu *et al.* 2018) and others (Tobin *et al.* 2010) have shown that oral glucose loading stimulates adipose tissue MBF in healthy people, which provides one potential explanation. Taken together, our study demonstrates that the route of glucose administration is a key determinant of postprandial MBF responses and provides a novel mechanistic explanation for the divergent MBF responses of oral *versus* intravenously administered glucose.

As expected, we observed marked differences in plasma insulin concentrations between challenges. Although the 'incretin effect' provides an explanation for the differences in insulin concentrations (Kazakos, 2011), the greater levels of plasma insulin (a vasodilator in healthy humans) following the OGTT did not result in augmented MBF. These data provide compelling evidence of alternative factors in the regulation of MBF following oral glucose

loading. However, measuring the skeletal muscle interstitial glucose and insulin concentrations to determine the impact of MBF on their delivery to the myocyte, and the rates of glucose disposal under experimental conditions where both glucose and insulin excursions are matched are important to follow up.

Considering the previously reported role of incretins (specifically GLP-1) in regulating skeletal muscle MBF (Sjøberg *et al.* 2013; Subaran *et al.* 2014; Tan *et al.* 2018), we investigated whether incretins, or other gut-derived factors, could explain the divergent muscle MBF responses observed between oral and intravenous glucose loading. GLP-1 has been reported to recruit capillaries in the skeletal muscle of animals (Sjøberg *et al.* 2013; Chai *et al.* 2014) and humans (Sjøberg *et al.* 2013; Subaran *et al.* 2014; Tan *et al.* 2018). Our data show a significant but transient increase in total and active plasma GLP-1 during the OGTT. However, neither active nor total GLP-1 concentrations were associated with changes

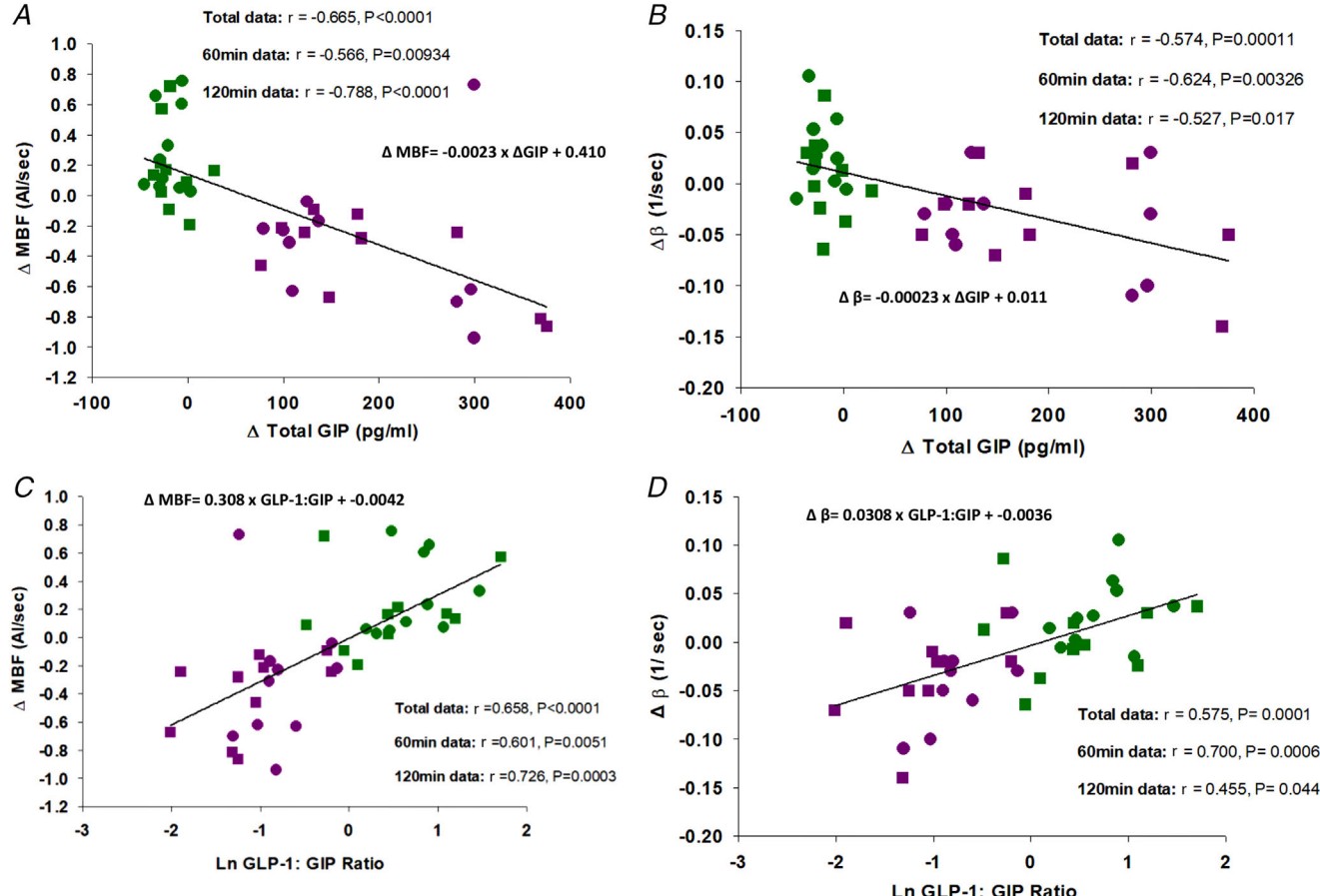

**Figure 4. Associations between skeletal muscle microvascular haemodynamics and plasma gut-derived hormone concentrations**
Graphs show OGTT (magenta), IVGTT (green), 60 min (circles) and 120 min (squares) data. Changes (Δ) in MBF *versus* Δ total GIP (*A*), Δβ *versus* Δ total GIP (*B*), ΔMBF *versus* natural log total GLP-1:GIP (*C*) and Δβ *versus* natural log total GLP-1:GIP (*D*). Continuous line and equation represent polynomial linear line of best fit for total data set. [Colour figure can be viewed at wileyonlinelibrary.com]

in muscle MBF. The potential vasodilatory actions of GLP-1 may have been 'masked' by the higher GIP concentrations during the OGTT, which we now hypothesise to be vasoconstrictive in skeletal muscle. In the current study, we report a strong negative association between plasma GIP concentrations and muscle MBF. Whilst GIP has been shown to play an active role in augmenting blood flow and microvascular responses in adipose tissue (Asmar *et al.* 2016, 2019), the vascular actions of GIP in skeletal muscle have not previously been investigated. Although our findings of GIP impairing MBF may at first appear to be contradictory (Asmar *et al.* 2016, 2019), the regulation of blood flow in adipose tissue is believed to be largely dominated by adrenergic mechanisms, which differs from that of skeletal muscle (Frayn & Karpe, 2014; Asmar *et al.* 2019). As such, we then assessed the relationship between the ratio of GLP-1:GIP and muscle MBF and demonstrated a strong positive relationship. Although this is not a causal relationship, it does highlight that the ratio of these incretins may also play an important role in postprandial skeletal muscle MBF. Follow-up studies to determine the direct effect of GIP alone, and in combination with GLP-1, are now required. Taken together, these data are suggestive of a novel incretin-mediated mechanism to skeletal muscle MBF in response to oral glucose loading.

Elevated GIP levels have been implicated in obesity and glucose intolerance in humans (Creutzfeldt *et al.* 1978; Góralska *et al.* 2020). Interestingly, mice lacking the GIP receptor are protected against high fat diet-induced obesity and insulin resistance (Miyawaki *et al.* 2002). In addition to GLP-1 and GIP receptors being located in the pancreas (Dillon *et al.* 1993; Gremlich *et al.* 1995), they are also located on vascular endothelial cells (Lim *et al.* 2017). GLP-1 is associated with NO production during hyperglycaemia in cultured endothelial cells (Lim *et al.* 2017) and stimulates skeletal muscle MBF in healthy rats via a NO synthase-dependent pathway (Dong *et al.* 2013). The mechanism by which GIP impairs skeletal muscle MBF is unknown; however, there is evidence that it stimulates release of endothelin-1 (ET-1; a potent vasoconstrictor) in cultured arterial endothelial cells (Ding *et al.* 2004). This may provide some potential clues to the mechanism of GIP vasoconstriction given that ET-1 opposes insulin-stimulated MBF in rats (Ross *et al.* 2007), which is NO-dependent (Vincent *et al.* 2004). This mechanism is speculative, but if ET-1 is involved, the co-infusion of an ET-1 receptor antagonist with GIP in humans could help answer this.

In the current study, plasma ghrelin and PYY were examined as potential gut-derived links to muscle MBF. Whilst we are not aware of any associations between PYY and skeletal muscle MBF, there is some evidence for PYY in the increase of blood flow and reduction of vascular resistance in parts of the brain (Tuor *et al.* 1988). Moreover, some positive associations have been drawn between ghrelin and increased perfusion of central organs following sepsis (Wu *et al.* 2005) and ischaemia–reperfusion injury (Bukowczan *et al.* 2015). However, we did not detect any associations between PYY or ghrelin and skeletal muscle MBF in our experimental protocol, and therefore propose that ghrelin and PYY are unlikely to be involved in the impaired MBF response during the OGTT.

Several limitations were identified in this study. First, to closely match blood glucose levels between the conditions, the OGTT had to be performed first and thus randomisation of the testing conditions was unable to be performed. Therefore, we cannot rule out an order effect of the testing conditions. Second, the measurement of gut-derived hormones in plasma was a targeted approach, and therefore we cannot rule out the involvement of other (or unknown) gut-derived factors which may have contributed to our observed blood flow effects. Third, we acknowledge that our participants were a healthy and young population and therefore cannot assume that MBF and gut hormone responses would be comparable in overweight individuals or those with type 2 diabetes. Finally, our study exclusively used contrast-enhanced ultrasound to measure microvascular responses in skeletal muscle. The use of other non-invasive microvascular perfusion techniques in parallel with contrast-enhanced ultrasound would corroborate our findings.

In conclusion, we have directly shown for the first time that an OGTT leads to decreased skeletal muscle MBF in healthy people, whereas a variable rate IVGTT matched for blood glucose excursions leads to increased skeletal muscle MBF. We uncovered novel associations between plasma GIP concentrations and GLP-1:GIP ratio and MBF responses, suggesting that glucose ingestion leads to secretion of GIP from the gut, which may promote vasoconstriction in the skeletal muscle microvasculature. Further work focused on the direct effects of a GIP infusion on the microvasculature is required to better understand this relationship.

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

## Additional information

### Data availability statement

Data generated for the current study are available on reasonable request from the corresponding author in the form of excel spreadsheets.

### Competing interests

None.

### Author contributions

K.M.R-T. and M.A.K. were responsible for the conception and design of the research. K.M.R-T. and M.A.K. performed

all experiments. L.P., A.C.B. and P.A.D.G. assisted in the data collection. K.M.R-T. performed all statistical analyses. All authors interpreted the data. K.M.R-T. drafted the manuscript and M.A.K. provided first edits; all authors revised the manuscript. All authors have read and approved the final version of this manuscript and agree to be accountable for all aspects of the work in ensuring that questions related to the accuracy or integrity of any part of the work are appropriately investigated and resolved. All persons designated as authors qualify for authorship, and all those who qualify for authorship are listed.

## Funding

L.P. is supported by a NHMRC & National Heart Foundation Fellowship (APP1157930). This work was supported by the Diabetes Australia Research Program (Y21G-KESM) and the Deakin University Institute for Physical Activity and Nutrition (2020-2021) seed funds.

## Acknowledgement

Open access publishing facilitated by Deakin University, as part of the Wiley – Deakin University agreement via the Council of Australian University Librarians.

## Keywords

cardiovascular physiology, gastrointestinal physiology, hyper-glycaemia, insulin, skeletal muscle

## Supporting information

Additional supporting information can be found online in the Supporting Information section at the end of the HTML view of the article. Supporting information files available:

**Peer Review History**
**Statistical Summary Document**

