## [Peer Review History · The Journal of Physiology]

Oral and intravenous glucose administration elicit opposing microvascular blood flow responses in skeletal muscle of healthy people: role of incretins

Katherine Roberts-Thomson, Lewan Parker, Andrew C Betik, Glenn D Wadley, Paul Della-Gatta, Thomas H Marwick, and Michelle A Keske

DOI: 10.1113/JP282428

Corresponding author(s): Michelle Keske (Michelle.Keske@deakin.edu.au)

Review Timeline:

Submission Date:	14-Oct-2021
Editorial Decision:	17-Nov-2021
Revision Received:	14-Dec-2021
Accepted:	11-Jan-2022

Senior Editor: Kim Barrett

Reviewing Editor: Bettina Mittendorfer

Transaction Report:

Dear Dr Keske,

Re: JP-RP-2021-282428 "Oral and intravenous glucose administration elicit opposing microvascular blood flow responses in skeletal muscle of healthy people: role of incretins" by Katherine Roberts-Thomson, Lewan Parker, Andrew C Betik, Glenn D Wadley, Paul Della-Gatta, Thomas H Marwick, and Michelle A Keske

Thank you for submitting your manuscript to The Journal of Physiology. It has been assessed by a Reviewing Editor and by 2 expert Referees and I am pleased to tell you that it is considered to be acceptable for publication following satisfactory revision.

The reports are copied at the end of this email. Please address all of the points and incorporate all requested revisions, or explain in your Response to Referees why a change has not been made.

NEW POLICY: In order to improve the transparency of its peer review process The Journal of Physiology publishes online as supporting information the peer review history of all articles accepted for publication. Readers will have access to decision letters, including all Editors' comments and referee reports, for each version of the manuscript and any author responses to peer review comments. Referees can decide whether or not they wish to be named on the peer review history document.

Authors are asked to use The Journal's premium BioRender (<https://biorender.com/>) account to create/redrawn their Abstract Figures. Information on how to access The Journal's premium BioRender account is here: <https://physoc.onlinelibrary.wiley.com/journal/14697793/biorender-access> and authors are expected to use this service. This will enable Authors to download high-resolution versions of their figures.

I hope you will find the comments helpful and have no difficulty returning your revisions within 4 weeks.

Your revised manuscript should be submitted online using the links in Author Tasks Link Not Available.

Any image files uploaded with the previous version are retained on the system. Please ensure you replace or remove all files that have been revised.

REVISION CHECKLIST:

- Article file, including any tables and figure legends, must be in an editable format (eg Word)
- Abstract figure file (see above)
- Statistical Summary Document
- Upload each figure as a separate high quality file
- Upload a full Response to Referees, including a response to any Senior and Reviewing Editor Comments;
- Upload a copy of the manuscript with the changes highlighted.

- A potential 'Cover Art' file for consideration as the Issue's cover image;
- Appropriate Supporting Information (Video, audio or data set https://jp.msubmit.net/cgi-bin/main.plex?form_type=display_requirements#supp).

To create your 'Response to Referees' copy all the reports, including any comments from the Senior and Reviewing Editors, into a Word, or similar, file and respond to each point in colour or CAPITALS and upload this when you submit your revision.

I look forward to receiving your revised submission.

If you have any queries please reply to this email and staff will be happy to assist.

Yours sincerely,

Professor Kim E. Barrett
Editor-in-Chief
The Journal of Physiology
<https://jp.msubmit.net>
<http://jp.physoc.org>
The Physiological Society
Hodgkin Huxley House
30 Farringdon Lane
London, EC1R 3AW
UK
<http://www.physoc.org>
<http://journals.physoc.org>

EDITOR COMMENTS

Reviewing Editor:

The reviewers found considerable merit in the paper and provided constructive feedback that will help to further improve this already strong manuscript.

REFEREE COMMENTS

Referee #1:

The authors compared the effects glucose ingestion and isoglycemic intravenous glucose infusion on skeletal muscle microvascular blood flow. They found, glucose infusion, but not glucose ingestion, caused an increase in microvascular blood flow. The mechanisms responsible for this effect are unknown, because the increase in plasma insulin, which is well-known to increase perfusion, was greater after glucose ingestion than glucose infusion. The authors speculate the difference could be related to incretins. However, there is little support for this from the literature because GLP-1 increases skeletal muscle microvascular perfusion. Nevertheless, the data are intriguing and clearly demonstrate that the route of glucose administration is a key determinant of postprandial skeletal muscle microvascular blood flow. The results from this study have important implications for the study of blood flow regulation and how it might be altered in people with obesity and diabetes or other conditions.

The paper is very well written and clear. I only have a few minor comments the authors may want to consider.

1) What was the total infused glucose load during the intravenous glucose infusion protocol and how does it compare to the

75 g of glucose provided during the glucose ingestion protocol.

2) What was the impact of the observed differences in perfusion on the insulin and glucose delivery rates to skeletal muscles during the oral and intravenous tests.

3) Adding glucagon concentrations would be informative.

Referee #2:

This is a neatly designed study that tests the hypothesis that oral glucose elicits different vascular response compared to intravenous glucose. This is an important first observation that points towards future studies on the impact of incretins and muscle microvascular blood flow regulation. While the study is explorative and only identify potential relationships, it has its justification from highlighting new avenues to be taken in the glucose-blood flow regulation research.

While a major limitation of the study is the non- mechanistic approach, I recognize that these observations of apparent relationships are important.

My minor comments are:

The number of figures should be limited. A total of 33 panels, in addition to the tables is overwhelming and does not improve the readability of the paper. Are all delta figures needed? Fig3. What does femoral artery diameter and velocity add to the story? Absolut flow makes sense to report but would you expect the others to be affected by the interventions? (and is the operator able to detect the 0.01-0.02 cm changes?)

If the individual data are presented, together with the box plot, it would make sense to have a line between the time points. I.e. to be able to actually see what the individual changes with the interventions were. With just the circles, the reader are left with less than nothing in terms of insight into the actual data set (fig 2 and 3).

The rationale for studying PYY and ghrelin should be improved in the intro or discussion.

Statistics: When and why were ANOVA vs. mixed model used? Some data points were missing. Please make clear when and where.

What was the study powered to detect? And based on what previous data? With the SD reported, it appears likely that the study was underpowered to detect differences in delta changes between the two interventions (knowing the variation usually observed in CEUS and doppler flow).

In #2 in the opening of the discussion: "but likely linked to the gut". This argument appears unsupported in its present form since the arguments comes in the following points. Maybe just omit from here as this is not what is shown but what is concluded based on the observations.

In the discussion, please elaborate on the potential mechanism by which the incretins (may) affect microvascular regulation. And, maybe also include a perspective as to how these potential mechanisms should be tested in a future study.

In the limitation, I don't see it as a major limitation that it was not possible to randomize the protocols. That is a given, with the used methodology. However, I would suggest to include some comments on the limitations of the CEUS method as this is a method that is often discussed/questioned.

END OF COMMENTS

Confidential Review

14-Oct-2021

14th December, 2021

Editor-in-Chief, The Journal of Physiology: Professor Kim E. Barrett

Dear Professor Barrett,

We would like to thank the reviewers for their careful and thoughtful review of our manuscript entitled “*Oral and intravenous glucose administration elicit opposing microvascular blood flow responses in skeletal muscle of healthy people: role of incretins*”. We appreciate their helpful and constructive comments and suggestions. Our comments and responses to the reviewer’s recommendations are as follows:

Reviewing Editor:

The reviewers found considerable merit in the paper and provided constructive feedback that will help to further improve this already strong manuscript.

Thank you. We have taken on board the reviewer’s suggestions and feedback.

Reviewer #1:

The authors compared the effects glucose ingestion and isoglycemic intravenous glucose infusion on skeletal muscle microvascular blood flow. They found, glucose infusion, but not glucose ingestion, caused an increase in microvascular blood flow. The mechanisms responsible for this effect are unknown, because the increase in plasma insulin, which is well-known to increase perfusion, was greater after glucose ingestion than glucose infusion. The authors speculate the difference could be related to incretins. However, there is little support for this from the literature because GLP-1 increases skeletal muscle microvascular perfusion. Nevertheless, the data are intriguing and clearly demonstrate that the route of glucose administration is a key determinant of postprandial skeletal muscle microvascular blood flow. The results from this study have important implications for the study of blood flow regulation and how it might be altered in people with obesity and diabetes or other conditions.

The paper is very well written and clear. I only have a few minor comments the authors may want to consider.

1) What was the total infused glucose load during the intravenous glucose infusion protocol and how does it compare to the 75 g of glucose provided during the glucose ingestion protocol.

The total infused glucose load during the intravenous glucose infusion protocol was reported in Table 1 (Glucose infusion rates over the 120 minute IVGTT). The total amount of glucose infused was 29.9 ± 17.6 g, and therefore was lower than the 75g oral glucose load provided during the glucose ingestion protocol.

Changes to manuscript: None

2) What was the impact of the observed differences in perfusion on the insulin and glucose delivery rates to skeletal muscles during the oral and intravenous tests?

This is an interesting question. We assume the reviewer is referring to the rates of insulin and glucose delivery to the myocyte (i.e. in the interstitial space) and the impact of this on glucose disposal. Interstitial delivery rates (measured via microdialysis) and muscle glucose disposal (measured via infusion of isotopic glucose tracers or sampling blood from both the

femoral artery and vein) were not performed in this study. To answer that question would require a different experimental design where both glucose and insulin time courses are matched (in our design, the OGTT elicited significantly higher insulin excursions than the IVGTT). However, we agree that these findings are important to follow-up in future studies. A fundamental first step (this study) was to determine the microvascular responses to an OGTT vs IVGTT matched for glucose alone.

Changes to manuscript: We have added the following to the Discussion on page 15 paragraph 1.

“However, measuring the skeletal muscle interstitial glucose and insulin concentrations to determine the impact of MBF on their delivery to the myocyte, and the rates of glucose disposal under experimental conditions where both glucose and insulin excursions are matched, are important to follow up.”

3) Adding glucagon concentrations would be informative.

The glucagon time course and area under the curve data were reported in Figure 1; panel G and H.

Changes to manuscript: None

Reviewer #2:

This is a neatly designed study that tests the hypothesis that oral glucose elicits different vascular response compared to intravenous glucose. This is an important first observation that points towards future studies on the impact of incretins and muscle microvascular blood flow regulation. While the study is explorative and only identifies potential relationships, it has its justification from highlighting new avenues to be taken in the glucose-blood flow regulation research.

While a major limitation of the study is the non-mechanistic approach, I recognize that these observations of apparent relationships are important.

My minor comments are:

1) The number of figures should be limited. A total of 33 panels, in addition to the tables is overwhelming and does not improve the readability of the paper. Are all delta figures needed? Fig3. What does femoral artery diameter and velocity add to the story? Absolute flow makes sense to report but would you expect the others to be affected by the interventions? (and is the operator able to detect the 0.01-0.02 cm changes?)

Thank you for this suggestion. We have removed Figure 3 (femoral artery diameter, velocity and blood flow – which constituted 6 panels) as these data did not report any major findings.

Changes to manuscript: We have deleted Figure 3 (femoral artery data) and have added the following sentence to Results on page 11 paragraph 3:

“No changes in superficial femoral artery diameter, velocity or blood flow were detected (data not shown).”

2) If the individual data are presented, together with the box plot, it would make sense to have a line between the time points. I.e. to be able to actually see what the individual changes with the interventions were. With just the circles, the reader are left with less than nothing in terms of insight into the actual data set (fig 2 and 3).

Thank you for this suggestion. We have considered these changes (see graphs below), however, we feel the lines between individual data points (with a box and whisker plot) makes the graph difficult to interpret. We have kept the original graphs in the manuscript, however, if the reviewer and editor would prefer these graphs be replaced we would be happy to accommodate this change.

Changes to manuscript: None

Figure 2

3) The rationale for studying PYY and ghrelin should be improved in the intro or discussion.

Thank you for this suggestion. There are no data regarding the involvement of PYY and/or ghrelin in the regulation of skeletal muscle MBF. However, we explored a range of gut-derived factors as we hypothesised a novel link between the gut and the impaired MBF responses observed during the OGTT and we did not want to restrict our hormone targets to incretins alone. PYY and ghrelin have been linked to blood flow in the brain as well as central organs, however, no associations were found between either PYY or ghrelin and skeletal muscle MBF in our study. Although these were null findings, they are important to report to help direct future research in this area.

Changes to manuscript: We have added the following to the Discussion on page 16 paragraph 2:

“In the current study, plasma ghrelin and PYY were examined as potential gut-derived links to muscle MBF. Whilst we are not aware of any associations between PYY and skeletal muscle MBF, there is some evidence for PYY in the increase of blood flow and reduction of vascular resistance in the brain (Tuor et al., 1988). Moreover, some positive associations have been drawn between ghrelin and increased microvascular perfusion of central organs following sepsis (Wu et al., 2005) and ischemic/re-perfusion injury (Bukowczan et al., 2015). However, we did not detect any associations between PYY or ghrelin and skeletal muscle MBF in our experimental paradigm, and therefore propose that ghrelin and PYY are unlikely to be involved in the impaired MBF response during the OGTT.”

4) Statistics: When and why were ANOVA vs. mixed model used? Some data points were missing. Please make clear when and where.

Two-way repeated measures of ANOVA were used to compare all multiple means with time (at baseline and throughout the 120 min OGTT/IVGTT time course), unless data points were missing. Data was lost for one participant’s central and peripheral hemodynamic measures (Table 3, where n=9) due to equipment malfunction of the mobil-o-graph. A mixed model was used as the statistical test for these data as a two-way repeated measures of ANOVA could not be used. No other data are missing. Graphpad Prism uses a mixed effects model approach that gives the same results as repeated measures ANOVA if there are no missing values, and comparable results when there are missing values.

Changes to manuscript: The following sentence was removed from the Statistical Analysis section of the methods on page 9, paragraph 1:

“If data were missing, the participant for that particular measurement was omitted.”

The following sentence was added to the Statistical Analysis section of the methods on page 9, paragraph 1:

“Mixed model analyses were used for statistical testing of central and peripheral hemodynamic measures due to missing data for one participant.”

5) What was the study powered to detect? And based on what previous data? With the SD reported, it appears likely that the study was underpowered to detect differences in delta changes between the two interventions (knowing the variation usually observed in CEUS and Doppler flow).

Based on a *a priori* sample size calculation, it was estimated that 13 participants would be required to detect a two-fold difference in MBF between the OGTT and IVGTT ($\alpha = 0.05$, power = 80%). This was based on our previous work showing a two-fold difference in MBF following a 50g oral glucose load versus a mixed nutrient meal in healthy participants (Russell et al 2018 American Journal of Physiology 315, E307-E315). We hypothesised the OGTT versus IVGTT would produce a similar magnitude change in MBF. However, due to extended COVID-19 clinic closures (March 2020-October 2021) we were forced to cease recruitment once 10 participants had been tested. Based on the data we collected in our 10 participants, we have 99% power to detect Δ MBF during the OGTT (-0.31 ± 0.46 AI/sec) versus IVGTT (0.29 ± 0.28 AI/sec) at 60 mins ($\alpha=0.05$, effect size 1.49) - which is our most variable time point - and a power of 91% adjusting for multiple comparisons ($\alpha= 0.0125$). Therefore we are adequately powered with a sample size of 10, and we do not believe that recruiting three additional participants would change the results of the current study.

Changes to manuscript: We have added the following to the Methods on page 6 paragraph 1:

“Based on a priori sample size calculation, it was estimated that 13 participants would be required to detect a 2-fold difference in MBF between the OGTT and IVGTT (SD = 115%, effect size = 0.87, $\alpha = 0.05$, power = 80%) based on previous work (Russell et al., 2018). Ten participants completed the study between September 2019 and March 2020 prior to COVID-19 related clinic closures.”

6) In #2 in the opening of the discussion: "but likely linked to the gut". This argument appears unsupported in its present form since the arguments comes in the following points. Maybe just omit from here as this is not what is shown but what is concluded based on the observations.

Thank you for this suggestion. This change has been made.

Changes to manuscript: We have omitted the second half of #2 on page 14 paragraph 1. It now reads as follows:

“....2) these opposing vascular actions are not related to hyperglycemia per se;”

7) In the discussion, please elaborate on the potential mechanism by which the incretins (may) affect microvascular regulation. And, maybe also include a perspective as to how these potential mechanisms should be tested in a future study.

Thank you for this suggestion. We have added a paragraph on mechanisms to page 16 paragraph 1 in the Discussion:

“Elevated GIP levels have been implicated in obesity and glucose intolerance in humans (Creutzfeldt et al., 1978; Góralaska et al., 2020). Interestingly, mice lacking the GIP receptor protect them against high fat diet-induced insulin resistance (Miyawaki et al., 2002). In addition to GLP-1 and GIP receptors being located in the pancreas (Dillon et al., 1993; Gremlich et al., 1995), they are also located on vascular endothelial cells (Lim et al., 2017). GLP-1 is associated with NO production during hyperglycemia in cultured endothelial cells (Lim et al., 2017) and stimulates skeletal muscle MBF in healthy rats via a NO synthase-dependent pathway (Dong et al., 2013). The mechanism by which GIP impairs skeletal muscle MBF is unknown, however, there is evidence that it stimulates release of endothelin-1 (ET-1; a potent vasoconstrictor) in cultured arterial endothelial cells (Ding et al., 2004). This may provide some potential clues to the mechanism of GIP vasoconstriction given that ET-1 opposes insulin-stimulated MBF in rats (Ross et al., 2007), which is NO-dependent (Vincent et al., 2004). This mechanism is speculative, however, if ET-1 is involved, the co-infusion of an ET-1 receptor antagonist with GIP in humans could help answer this.”

8) In the limitation, I don't see it as a major limitation that it was not possible to randomize the protocols. That is a given, with the used methodology. However, I would suggest to include some comments on the limitations of the CEUS method as this is a method that is often discussed/questioned.

Thank you for this suggestion. These additions have been included in the manuscript.

Changes to manuscript: We have included the following in the Discussion on page 17 paragraph 1:

“Finally, our study exclusively used CEU to measure microvascular responses in skeletal muscle. The use of other non-invasive microvascular perfusion techniques in parallel with CEU would corroborate our findings.”

Other changes

The authors have completed the Statistical Summary Document and have identified some minor errors in data entry and statistical reporting which have now been amended (detailed below). These changes are minor and do not alter the main findings or the interpretation of the manuscript.

1) Table 3: Mean arterial pressure

There was one data entry error for MAP at 0 min in the IVGTT condition. This has been changed from “ 95 ± 11 mmHg” to “ 97 ± 12 mmHg”.

2) Table 4: Active Ghrelin and PYY (OGTT - Δ from 0 min)

The change in active ghrelin between the OGTT and IVGTT was not significant (no main effect or interaction) and therefore the # symbols have been removed from Table 4.

The change in PYY was higher for the OGTT compared to the IVGTT at all time points. Symbols (#) have been added to these time points in Table 4. The following sentence was added to the Results on page 12, paragraph 5 “ Δ PYY was significantly higher at all time points during the OGTT compared to the IVGTT (Table 4)”

References

- Bukowczan J, Warzecha Z, Ceranowicz P, Kusnierz-Cabala B, Tomaszewska R & Dembinski A. (2015). Therapeutic effect of ghrelin in the course of ischemia/reperfusion-induced acute pancreatitis. *Current Pharmaceutical Design* **21**, 2284-2290.
- Creutzfeldt W, Ebert R, Willms B, Frerichs H & Brown J. (1978). Gastric inhibitory polypeptide (GIP) and insulin in obesity: increased response to stimulation and defective feedback control of serum levels. *Diabetologia* **14**, 15-24.
- Dillon JS, Tanizawa Y, Wheeler M, Leng X-H, Ligon BB, Rabin D, Yoo-Warren H, Permutt M & Boyd 3rd A. (1993). Cloning and functional expression of the human glucagon-like peptide-1 (GLP-1) receptor. *Endocrinology* **133**, 1907-1910.
- Ding K-H, Zhong Q, Xu J & Isales CM. (2004). Glucose-dependent insulinotropic peptide: differential effects on hepatic artery vs. portal vein endothelial cells. *American Journal of Physiology-Endocrinology and Metabolism* **286**, E773-E779.
- Dong Z, Chai W, Wang W, Zhao L, Fu Z, Cao W & Liu Z. (2013). Protein kinase A mediates glucagon-like peptide 1-induced nitric oxide production and muscle microvascular recruitment. *American Journal of Physiology-Endocrinology and Metabolism* **304**, E222-E228.
- Góralaska J, Rażny U, Polus A, Dziewońska A, Gruca A, Zdzienicka A, Dembińska-Kieć A, Solnica B, Micek A & Kapusta M. (2020). Enhanced GIP Secretion in Obesity Is Associated with Biochemical Alteration and miRNA Contribution to the Development of Liver Steatosis. *Nutrients* **12**, 476.
- Gremlich S, Porret A, Cherif D, Vionnet N, Froguel P & Thorens B. (1995). Cloning, functional expression, and chromosomal localization of the human pancreatic islet glucose-dependent insulinotropic polypeptide receptor. *Diabetes* **44**, 1202-1208.
- Lim DM, Park KY, Hwang WM, Kim JY & Kim BJ. (2017). Difference in protective effects of GIP and GLP-1 on endothelial cells according to cyclic adenosine monophosphate response. *Experimental and therapeutic medicine* **13**, 2558-2564.
- Miyawaki K, Yamada Y, Ban N, Ihara Y, Tsukiyama K, Zhou H, Fujimoto S, Oku A, Tsuda K & Toyokuni S. (2002). Inhibition of gastric inhibitory polypeptide signaling prevents obesity. *Nature medicine* **8**, 738-742.
- Ross R, Kolka C, Rattigan S & Clark M. (2007). Acute blockade by endothelin-1 of haemodynamic insulin action in rats. *Diabetologia* **50**, 443-451.

- Russell RD, Hu D, Greenaway T, Sharman JE, Rattigan S, Richards SM & Keske MA. (2018). Oral Glucose Challenge Impairs Skeletal Muscle Microvascular Blood Flow in Healthy People. *American Journal of Physiology-Endocrinology and Metabolism* **315**, E307-E315.
- Tuor U, Kondysar M & Harding R. (1988). Effect of angiotensin II and peptide YY on cerebral and circumventricular blood flow. *Peptides* **9**, 141-149.
- Vincent MA, Clerk LH, Lindner JR, Klibanov AL, Clark MG, Rattigan S & Barrett EJ. (2004). Microvascular recruitment is an early insulin effect that regulates skeletal muscle glucose uptake in vivo. *Diabetes* **53**, 1418-1423.
- Wu R, Dong W, Zhou M, Cui X, Hank Simms H & Wang P. (2005). Ghrelin improves tissue perfusion in severe sepsis via downregulation of endothelin-1. *Cardiovascular research* **68**, 318-326.

Dear Dr Keske,

Re: JP-RP-2021-282428R1 "Oral and intravenous glucose administration elicit opposing microvascular blood flow responses in skeletal muscle of healthy people: role of incretins" by Katherine Roberts-Thomson, Lewan Parker, Andrew C Betik, Glenn D Wadley, Paul Della-Gatta, Thomas H Marwick, and Michelle A Keske

I am pleased to tell you that your paper has been accepted for publication in The Journal of Physiology.

Methods: Please could authors add a statement on database registration/Clause 35 (or else state "except for registration in a database").

NEW POLICY: In order to improve the transparency of its peer review process The Journal of Physiology publishes online as supporting information the peer review history of all articles accepted for publication. Readers will have access to decision letters, including all Editors' comments and referee reports, for each version of the manuscript and any author responses to peer review comments. Referees can decide whether or not they wish to be named on the peer review history document.

Are you on Twitter? Once your paper is online, why not share your achievement with your followers. Please tag The Journal (@jphysiol) in any tweets and we will share your accepted paper with our 23,000+ followers!

The last Word version of the paper submitted will be used by the Production Editors to prepare your proof. When this is ready you will receive an email containing a link to Wiley's Online Proofing System. The proof should be checked and corrected as quickly as possible.

Authors should note that it is too late at this point to offer corrections prior to proofing. The accepted version will be published online, ahead of the copy edited and typeset version being made available. Major corrections at proof stage, such as changes to figures, will be referred to the Reviewing Editor for approval before they can be incorporated. Only minor changes, such as to style and consistency, should be made a proof stage. Changes that need to be made after proof stage will usually require a formal correction notice.

All queries at proof stage should be sent to TJP@wiley.com

Yours sincerely,

Professor Kim E. Barrett
Editor-in-Chief
The Journal of Physiology
<https://jp.msubmit.net>
<http://jp.physoc.org>
The Physiological Society
Hodgkin Huxley House
30 Farringdon Lane
London, EC1R 3AW
UK
<http://www.physoc.org>
<http://journals.physoc.org>

P.S. - You can help your research get the attention it deserves! Check out Wiley's free Promotion Guide for best-practice recommendations for promoting your work at www.wileyauthors.com/eoo/guide. And learn more about Wiley Editing Services which offers professional video, design, and writing services to create shareable video abstracts, infographics, conference posters, lay summaries, and research news stories for your research at www.wileyauthors.com/eoo/promotion.

* IMPORTANT NOTICE ABOUT OPEN ACCESS *

Information about Open Access policies can be found here <https://physoc.onlinelibrary.wiley.com/hub/access-policies>

To assist authors whose funding agencies mandate public access to published research findings sooner than 12 months after publication The Journal of Physiology allows authors to pay an open access (OA) fee to have their papers made freely available immediately on publication.

You will receive an email from Wiley with details on how to register or log-in to Wiley Authors Services where you will be able to place an OnlineOpen order.

You can check if your funder or institution has a Wiley Open Access Account here <https://authorservices.wiley.com/author-resources/Journal-Authors/licensing-and-open-access/open-access/author-compliance-tool.html>

Your article will be made Open Access upon publication, or as soon as payment is received.

If you wish to put your paper on an OA website such as PMC or UKPMC or your institutional repository within 12 months of publication you must pay the open access fee, which covers the cost of publication.

OnlineOpen articles are deposited in PubMed Central (PMC) and PMC mirror sites. Authors of OnlineOpen articles are permitted to post the final, published PDF of their article on a website, institutional repository, or other free public server, immediately on publication.

Note to NIH-funded authors: The Journal of Physiology is published on PMC 12 months after publication, NIH-funded authors DO NOT NEED to pay to publish and DO NOT NEED to post their accepted papers on PMC.

EDITOR COMMENTS

Reviewing Editor:

No further comments. Nice work.

REFEREE COMMENTS

Referee #1:

No further comments.

Referee #2:

No further comments.

END OF COMMENTS

1st Confidential Review

14-Dec-2021